# Role of Catechol-O-methyltransferase Val158Met Polymorphism on Transcranial Direct Current Stimulation in Swallowing

**DOI:** 10.3390/jpm12030488

**Published:** 2022-03-17

**Authors:** Hyemi Hwang, Yeonjae Han, Geun-Young Park, Soohwan Lee, Hae-Yeon Park, Sun Im

**Affiliations:** 1Department of Rehabilitation Medicine, Bucheon St. Mary’s Hospital, College of Medicine, The Catholic University of Korea, Seoul 14647, Korea; hyemi22333@naver.com (H.H.); bucheon2170@gmail.com (Y.H.); rootpmr@catholic.ac.kr (G.-Y.P.); rhrhtjsql@naver.com (S.L.); 2Department of Rehabilitation Medicine, Seoul St. Mary’s Hospital, College of Medicine, The Catholic University of Korea, Seoul 06591, Korea

**Keywords:** dysphagia, transcranial direct current stimulation, catechol-O-methyltransferase, genetic polymorphism

## Abstract

Transcranial direct current stimulation (tDCS) is one of the latest post-stroke dysphagia treatment modalities, and the effect of tDCS is known to be affected by various factors including genetic polymorphisms. However, the role of catechol-O-methyltransferase (COMT) polymorphisms on tDCS in swallowing is unclear. In this prospective pilot study, we aim to explore the effect of tDCS on the swallowing cortex and subsequent swallowing motor function according to COMT polymorphism. Twenty-four healthy participants received either anodal tDCS or sham mode tDCS on the mylohyoid motor cortex at random order, after inhibitory repetitive transcranial magnetic stimulation (rTMS) for preconditioning. The primary outcome was the changes of mylohyoid-motor-evoked potentials (MH-MEP) amplitude in each COMT polymorphism group, from the post-inhibitory rTMS baseline state to immediate, 30, and 60 min after tDCS. The secondary outcomes were the changes in swallowing function. The results showed that COMT Val/Val polymorphism showed improvement across time in the MH-MEP amplitudes and triggering time of swallowing after tDCS, whereas COMT Met carrier group did not show significant changes of MH-MEP or swallowing function across time. This therapeutic response variability of tDCS in the mylohyoid motor system according to COMT polymorphism support the importance of genetic analysis in individualized dysphagia treatment.

## 1. Introduction

Dysphagia is one of the most common complications after stroke, accounting for 19–81% of stroke patients [1]. Post-stroke dysphagia increases hospitalization period and mortality due to malnutrition itself and subsequent aspiration pneumonia. Various treatment modalities for dysphagia management have been established, including diet modification, postural and compensatory swallowing techniques, and neuromuscular electrical stimulation [2,3]. In recent years, non-invasive brain stimulation (NIBS) techniques such as repetitive transcranial magnetic stimulation (rTMS) [4] or transcranial direct current stimulation (tDCS) [5,6,7] have been introduced and widely studied with a fast accumulation of evidence. Among these two techniques, the former has recently shown consistently positive results in enhancing post-stroke swallowing recovery [8].

tDCS is one of the NIBS modalities that can affect cortical excitability through modulation of neural activity using weak electrical currents [9,10], and has shown effectiveness in post-stroke dysphagia treatment [10]. However, the effects of tDCS may vary depending on the attentive level of the participant, detection of the motor hot spot, and also on the intrinsic factors such as age, gender, cortical thickness, or genes [11,12]. Recent studies have revealed that several genetic polymorphisms may also contribute to variability in therapeutic response to neurostimulation. Polymorphism in some single nucleotide polymorphism (SNP)s, such as brain-derived neurotrophic factor (BDNF) Val66Met, or catechol-O-methyl transferase (COMT) Val158Met have turned out to be associated with varying outcomes with tDCS application in cognitive and mood disorder [13,14].

The close relationship between BDNF SNPs and the post-stroke outcome has been delineated in post-stroke motor recovery [15,16]. In parallel, those with Val/Val allele of BDNF *rs6265* manifested faster and greater improvement than Met carrier on post-stroke dysphagia [17]. In addition, BDNF Val66Val has shown a positive association with the neuroplasticity through rTMS in motor function [18] and swallowing [19]. BDNF also has shown to modulate the effects of tDCS during chronic stroke-induced aphasia treatment [20] and have a key role in mediating the beneficial effects of tDCS on explicit memory [21].

In addition to BDNF, dopamine is also related to post-stroke recovery. The dopaminergic system plays a role in motor function (nigrostriatal pathway), motivated behavior, mood homeostasis and reward circuitry (mesolimbic pathway), cognition (mesocortical pathway), and swallowing [22]. Due to the diverse functions of dopamine, many genetic polymorphisms related to dopamine regulation were investigated and various genetic polymorphisms were studied across many fields [23,24]. Among those, dopamine regulators such as COMT and dopamine receptor genes (DRD1, DRD2, and DRD3) have been further explored. COMT is known to influence working memory, motor learning, and swallowing [25], and *rs4532* of DRD1 is reported to affect post-stroke swallowing recovery in the elderly [26]. Other studies have also reported the close association of DRD2 with swallowing [27,28].

Due to its role in dopamine regulation, the effects of COMT polymorphism on NIBS (rTMS or tDCS) have been widely studied. However, many of the studies has been mostly conducted in the psychiatric field, such as schizophrenia, mood, and cognitive disorders [29,30,31,32,33]. Given the links between dopamine level and dysphagia, and the association of the COMT SNPs with motor recovery after a stroke, we considered the possibility that COMT Val158Met polymorphism would affect swallowing function and responsiveness to NIBS. Yet, whether COMT SNPs may affect swallowing response after tDCS has not been studied before.

Therefore, we conducted the prospective pilot study with healthy participants to explore whether an individual’s COMT polymorphism may influence the response to the tDCS on the mylohyoid motor cortex and the subsequent actual swallowing function after induction of a transient dysphagia state. We hypothesized that COMT Val158Met polymorphism affecting dopamine neurotransmission may influence the effect of tDCS on the swallowing cortex, and that this genetic variance may affect the treatment response after tDCS in clinical practice.

## 2. Materials and Methods

### 2.1. Participants and Study Design

Twenty-four healthy participants (mean age, 22.1 ± 3.3 years old; range, 19–29 years old; 13 males; 11 females) were recruited from January 2019 to July 2019. Participants were excluded if they belonged to any of the following criteria: previous history of epilepsy, previous brain surgery, use of medication which acts on the central nervous system, placement or implanted metal, cardiac pacemaker, tattoo on face, impossible to collect the blood sample, any previous swallowing problems, and the possibility of pregnancy. Written informed consent was obtained from each subject before the study. This study was ethically approved by the institutional review board of the Catholic university of Korea, Bucheon Saint Mary’s hospital (HC18TNSV0061).

This study was the prospective pilot study, and all participants followed the protocols as illustrated in Figure 1. We designed the experimental flow that all participants receive either the anodal tDCS or sham mode tDCS on the mylohyoid motor cortex at random order after undergoing inhibitory rTMS for preconditioning. All the participants received the anodal tDCS and sham mode tDCS on two separate visits. The order of anodal tDCS and sham mode was randomly distributed among the participants and the duration between the first and second visits was at least one week.

### 2.2. DNA Sampling and Genotyping 

At the first visit, all participants underwent DNA sampling to collect 2 mL of whole blood. The blood sample was labelled with each participant’s de-identified study number. DNA extraction and genotyping of COMT *rs4680* (Val/Val and Met carrier groups) were done in similar procedures with the authors’ previous studies [17,26]. 

### 2.3. Motor-Evoked Potential and Cortical Excitability

Motor-evoked potentials (MEPs) were obtained from the participants asked to sit on a chair with an armrest. Cortical stimulation was performed using a commercially available MagPro X100 magnetic stimulator (Magventure A/S DK-3520, Farum, Denmark) connected to a 75-mm figure-of-eight shaped stimulation coil (MCF-B70). MEPs of thenar muscle were obtained at the enrollment. The MEPs of mylohyoid muscle were obtained at enrollment and after inhibitory rTMS (baseline), immediately, 30 and 60 min after tDCS. The same time intervals were followed for the sham tDCS session (Figure 1).

Thenar MEPs were recorded from the abductor pollicis brevis (APB) muscle with the magnetic stimulation of contralateral primary motor cortex (M1), in compliance with the previously reported method [34]. The resting motor threshold (rMT) was defined as the lowest stimulus intensity, which could induce MEPs of at least 50 μV peak-to-peak amplitude in five out of ten consecutive trials. Five sweeps of the MEPs at 120% of the rMT were collected, and the mean amplitude and latency of the MEPs were calculated [34]. 

Mylohyoid MEPs (MH-MEPs) were obtained using the same methods as Hamdy et al. [9]. Bipolar silver-silver chloride electrodes were used, and they were positioned submentally, 2 cm lateral to the midline, one over the left and one over the right mylohyoid muscle. The reference electrode was positioned on the mentum of the mandible. To determine the optimal stimulation point (“hot spot”), the coil was tangentially positioned around the mylohyoid motor cortex, 2–4 cm anterior and 4–6 cm lateral to the vertex [9]. Mylohyoid rMT was obtained in the same way as APB rMT, and through five sweeps of the MH-MEPs at 120% of the mylohyoid rMT, the mean amplitude and latency of the MH-MEPs were calculated. The dominant mylohyoid hemisphere was defined as the hemisphere where cortical stimulations elicited the maximal MH-MEP amplitude, judged on the previous studies [9,35].

### 2.4. Inhibitory Repetitive Transcranial Magnetic Stimulation

MagPro X100 magnetic stimulator (Magventure A/S DK-3520, Farum, Denmark) was used for stimulation, and 1 Hz frequency of rTMS was delivered at 120% of the mylohyoid rMT. Two trains of 300 pulses with 30 s inter-train interval were delivered over the “hot-spot”, for 10 min over the dominant cortex.

### 2.5. Transcranial Direct Current Stimulation 

As described above (Figure 1), all the participants received both (1) anodal tDCS (1 mA for 20 min) or (2) sham mode tDCS (1 mA for 30 s) on a random sequence, which was generated by tDCS device (NeuroConn, Ilmenau, Germany). tDCS device was pre-programmed with 22 five-digit codes triggering anodal tDCS and 22 codes triggering sham mode tDCS. At the first visit, each participant was assigned to the random code (anodal or sham mode) generated by the tDCS device, and at the second visit, an unused tDCS code was assigned. tDCS stimulation was conducted by an investigator who did not participate in outcome measurements or data analysis.

The anodal tDCS or sham mode tDCS was delivered to the dominant hemisphere by a battery-driven, direct current stimulator through two sponge surface electrodes. Anodal electrode (5 × 7 cm^2^) was placed over the patient’s scalp of the dominant hemisphere, where maximal MH-MEPs was obtained, and a reference electrode (10 × 10 cm^2^) was placed over the contralateral supraorbital region [9,36,37]. The electrodes were secured using adjustable rubber straps placed around the head. In anodal tDCS mode, the direct current was increased to 1 mA incrementally over several seconds and was maintained for 20 min [38]. In sham mode tDCS, 1 mA current was delivered for only 30 s, producing an initial tingling sensation but no significant changes in cortical excitability [39]. 

### 2.6. Electrophyioslogical Evaluation of Oropharyngeal Swallowing Function

The evaluation of oropharyngeal swallowing function was performed via electrophysiological method previously described by Ertekin et al. [40,41,42]. Both submental-electromyography (SM-EMG) and laryngeal movements were recorded from the onset of swallowing a single bolus of water. During each assessment, 10 mL of water was placed in the participant’s mouth using a syringe, and the participant was instructed to swallow water as directed.

SM-EMG was performed using surface electrodes placed on suprahyoid muscles under the chin. The onset, total duration, and maximal amplitudes of SM-EMG were measured during the single bolus swallowing. Vertical upward and downward movements of the larynx were measured by a piezoelectric movement sensor (or laryngeal movement sensor) placed between the thyroid and cricoid cartilages at the midline. Two subsequent deflections of laryngeal movement sensor signals represent laryngeal upward and downward movements, respectively (Figure 2). The larynx and airway are closed from the onset of the first deflection until the onset of the second deflection, and the time between these two-time points is defined as the laryngeal relocation time. Triggering time, defined as the interval between the onset of SM-EMG and the onset of the first deflection of laryngeal signal, provides information on the temporal relationship between the voluntary activation of submental-muscle complex and reflex triggering of the swallowing response.

Oropharyngeal swallowing function as measured by SM-EMG was measured at enrollment, after inhibitory rTMS (baseline), and immediately, 30, and 60 min after tDCS in parallel with cortical excitability measurements.

### 2.7. Statistical Analysis

Sample size calculation was based on previous studies using the inhibitory pre-conditioning model as published previously [38,43,44], which revealed that 12 participants would be required to achieve a power of 80% and statistical significance of 5% (with a standard deviation of 2.5). Considering that two allele groups are compared (COMT Val/Val and Met carrier groups), 12 participants in each group, a total of 24 participants, were required.

All statistical analyses were performed using R Statistical Software (version 4.1.2; R Foundation for Statistical Computing, Vienna, Austria). Assumptions of a normal distribution of the outcome variables were assessed using the Shapiro–Wilk test. Due to the non-normality of the data, pairwise comparisons within and between groups were performed with the Wilcoxon signed-rank test or Wilcoxon rank test. To assess the serial changes across time of the response variables in a non-parametric manner, Friedman test was performed to compare the parameters between the Val/Val and Met carrier groups. *p* values of 0.05 were taken as a measure of statistical significance, and data are expressed as mean values with standard error of the mean (SEM) unless stated otherwise.

The primary outcome was the changes of MH-MEP amplitude in each COMT polymorphism group, from the post-inhibitory rTMS baseline state to immediate, 30, and 60 min after tDCS application. The secondary outcomes were the changes in swallowing function as reflected from the laryngeal sensors and SM-EMG. To minimize the effects of age and gender which may lead to inter-individual variations in the electrophysiological parameters, percent changes from the baseline (i.e., post-inhibitory rTMS state) were used for statistical analysis. Adjustment for multiple comparisons was performed using Bonferroni correction. When significant results across time were found, pairwise comparison at each measurement were performed using Wilcoxon singed rank sum test.

## 3. Results

### 3.1. Assessments at the Enrollment

Regarding COMT genotype, 12 patients were grouped into Val/Val (age, 20.4 ± 6.9 years old; male, 58.3%), and 12 patients were groups into Met carrier (age, 19.3 ± 3.2 years old; male 50.0%). Assessments of the participants according to genotype groups at the enrollment are shown in Table 1. No significant group differences were observed in all the assessments at enrollment. No drop-out was observed across the experimental procedures. The dominance of the swallowing cortex was observed on the right side in 15 of the participants. The mean APB rMT was 19.8 ± 5.2 mV.

### 3.2. Post-Inhibitory rTMS (Baseline Assessment)

Transient dysphagia was induced with rTMS inhibition, and both the tDCS and sham modes showed similar degree of inhibition. The inhibitory effects were reflected in the MH-MEP amplitudes (mean ± SD) changes, in the tDCS session (initial: 94.2 ± 17.8 mV versus inhibition: 80.9 ± 17.6 mV, *p*-value = 0.016) and sham session (initial: 104.6 ± 19.2 mV versus inhibition: 71.1 ± 14.6 mV, *p*-value = 0.022). These inhibitory effects after induction of dysphagia were also observed in the MH-MEP latencies (tDCS: *p*-value = 0.003, sham: *p*-value = 0.022). No significant differences were observed in the degree of cortical inhibition between the tDCS and sham sessions.

### 3.3. Dysphagia Reversal, tDCS Versus Sham Session

Compared to baseline, both the tDCS and sham mode sessions showed reversal of the inhibitory effects with increased amplitudes of MH-MEPs recorded 60 min after the intervention (Table 2). However, improvement in the electrophysiological swallowing function, as reflected in changes of laryngeal relocation and trigger time was only observed in the tDCS session. In addition, the tDCS led to greater degree of changes in cortical excitability than the sham mode (Table 2).

During the tDCS intervention, no adverse event was observed. None of the participants showed evidence of skin lesions, headaches, tingling sensation, or nausea after receiving the tDCS.

### 3.4. Effects of Genetic Polymorphism on Cortical Excitability after tDCS

The Val-Val group showed significant improvement in the MH-MEP amplitude percentage changes across time, from immediate to 60 min after tDCS (χ^2^ (2) = 11.17, *p*-value = 0.0038), confirmed by post-hoc Wilcoxon singed-rank test with Bonferroni correction (adjusted *p*-value = 0.028). In the Met carrier group, there was no statistical significance change in MH-MEP amplitude or latency changes across the follow-up period (Figure 3).

### 3.5. Effects of Genetic Polymorphism on Swallowing Function after tDCS

The Val/Val group showed statistically significant changes of the swallowing trigger time across time (χ^2^(2) = 6.68, *p*-value = 0.035), though this was not observed in the Met carrier group (χ^2^(2) = 4.17, *p*-value = 0.12). Post hoc Wilcoxon signed-rank test in Val/Val group showed statistical significance of percent change in the trigger time between immediate and 60 min after tDCS (adjusted *p*-value = 0.01) (Figure 4).

The two phenotypes showed no significant intergroup differences in the SM-EMG total duration and maximal amplitude and laryngeal relocation time across time.

## 4. Discussion

In this prospective pilot study with healthy participants, where inhibitory rTMS created transient unilateral lesion reflecting swallowing impairment, we explored whether COMT *rs4680* polymorphism influences tDCS response on the mylohyoid motor cortex. Compared with sham mode sessions, tDCS had a more significant impact on cortical excitability and swallowing function. Further classification according to COMT genotypes showed that Val/Val polymorphism showed significant improvement across time in the MH-MEP amplitudes and swallowing triggering time after tDCS.

Previous studies with the effects of tDCS on swallowing have also shown the improvement of MEP amplitudes over time. In one study with healthy volunteers, tDCS was applied over the hemisphere with both stronger and weaker suprahyoid/submental projections, and suprahyoid MEPs were assessed immediately before the intervention (baseline), as well as 5, 30, 60, and 90 min after the intervention. Post-tDCS increased suprahyoid MEP amplitudes from the stimulated hemispheres compared to the sham group up to 60 min after intervention in both hemispheres [45]. Another study has also shown increased pharyngeal MEP amplitudes up to 60 min after applying anodal tDCS for 20 min at 1 mA and 10 min at 1.5 mA [38]. Similarly, in our study, MH-MEP amplitudes, which reflect suprahyoid cortical excitability, showed better improvement after anodal tDCS compared with the sham group, specifically in the COMT Val/Val groups compared with the Met carrier group.

Triggering time, which represents the initiation of the oropharyngeal swallowing reflex, is important in evaluating swallowing disorder. Triggering time tends to increase with age and is significantly prolonged in stroke, Parkinson’s disease, amyotrophic lateral sclerosis, and pseudobulbar palsy [46]. Therefore, triggering time during a bolus test could indirectly give information on neurogenic dysphagia [46]. In our study, tDCS had a greater impact on reducing triggering time than sham sessions. In the case of COMT genotypes, Val/Val polymorphism showed significant reduction in triggering time after tDCS. From these results, we may assume the positive effects of tDCS on swallowing reflex, and that COMT polymorphism affects mylohyoid cortex excitability and actual swallowing function after anodal tDCS.

COMT is an enzyme that influences the availability of dopamine in the synaptic cleft by stimulating dopamine degradation, responsible for more than 60% of dopamine metabolic degradation in the frontal cortex [47]. It is highly expressed in the entire brain and plays an important role in dopamine flux inside the prefrontal cortex [48]. The Val allele is almost four times as active as the Met allele at normal body temperature in COMT Val158Met polymorphism. Therefore Val/Val allele carriers demonstrate the lowest dopamine level, whereas the Met/Met carriers demonstrate the highest dopamine level [49].

Contrary to the speculation that those with the Met allele with higher dopamine levels would show a better improvement in swallowing after tDCS, Val/Val carriers showed a better response in our study. Many of the previous studies of COMT with tDCS, mostly conducted in the psychiatric fields and cognition, have shown similar results [29,30,31,32,33]. In patients with schizophrenia, Val/Val allele was associated with significantly more reduction of auditory hallucinations following tDCS at the dorsolateral prefrontal cortex (DLPFC) compared to Met carriers [31]. Other studies have reported that Val carriers showed greater plasticity of working memory in healthy young participants between 18 and 35 years old [29,33]. Another study of healthy adults showed that COMT Met/Met group was found to have lower set-shifting ability after anodal tDCS to DLPFC than Val/Val carriers [30]. Moreover, in a study with healthy right-handed older adults (mean age: 69.03), the Val/Val group showed improvement in visual working memory and spatial working memory after receiving low to intermediate intensity anodal tDCS at the frontoparietal area, whereas Met/Met group showed decreased spatial working memory performance following high-intensity tDCS [32].

COMT gene is known to affect motor function in addition to cognition and memory. Because motor learning and motor control require diverse cognitive functions [50] and because the dopaminergic system is closely related to motor learning [24], COMT polymorphism is also expected to be associated with motor learning and recovery. In a recent study, Val/Val phenotype showed greater improvement in Fugl-Meyer Assessment and Functional Independence Measure score at 3 and 6 months after stroke, indicating that dopamine-related COMT polymorphism may affect motor recovery after stroke [24]. Another study also reported that the motor recovery and activity of daily livings showed more improvement during both initial and 6 months post-stroke in patients with Val/Val alleles than those with Met/Met alleles [51].

Dopamine function on the prefrontal cortex is reported to follow an “inverted U-shape”, where both deficient and excessive dopamine activity results in poor functional outcomes and optimal dopamine levels result in good functional outcomes [26,52,53]. After tDCS, dopamine signaling is expected to increase [54]. In this study, the Val/Val group with low dopamine levels at the baseline may have shifted toward optimal dopamine level after tDCS, leading to better results. On the other hand, high dopamine levels in Met carriers may be at risk of excessive dopamine levels after tDCS, resulting in no significant improvement. Moreover, the Met allele is associated with tonic dopamine transmission, favoring the stabilization and maintenance of relevant information in working memory. In contrast, the Val allele is associated with increased phasic dopamine transmission, which favors cognitive flexibility required in task demanding updating and manipulating information [55,56]. Likewise, this phasic dopamine transmission of the Val allele may have contributed to greater outcome changes after tDCS in this study.

As previously described, several studies that reported significant response in Val carrier than Met carriers after tDCS were mainly limited to the psychiatric fields and cognition [29,30,31,32]. To our knowledge, this is the first study on swallowing motor recovery after tDCS according to COMT polymorphism in healthy adults. Furthermore, this study included the electrophysiological evaluation of swallowing function using SM-EMG and laryngeal sensor, which reflect the activation of swallowing muscles and reflex triggering of the swallowing response. The results of our study were in parallel to the previous studies in the neurocognitive field where Val/Val group showed greater response after tDCS than Met carriers [30,32].

However, there are some limitations of the study. First, the number of participants was small, and the deviation of the results between individuals was large. Some swallowing performance parameters may not have reached statistical significance due to the large deviations in the data. Second, this study was conducted on healthy, young patients under 30-years old (mean age: 22.13 ± 3.26 years old; range: 19–29 years old). Therefore, comparison between the stroke group and non-stroke group, and the young group and the older group was not feasible. Nevertheless, because this study was confined to young and healthy participants, the results of our study can be interpreted by a single factor—COMT polymorphism. Though the methods used in this study closely resemble that from other previous studies [19,35], the clinical implications of our findings need to be confirmed through actual tDCS application in patients with post-stroke dysphagia.

Our study suggests that those with the COMT Val/Val allele may show better improvement after anodal tDCS than those with the Met-allele. Although further large-scale studies with more participants with neurogenic dysphagia are recommended to unravel the potential mechanisms of genetic influence on swallowing and cortical plasticity, this study provides insight into the understanding of response variability across COMT genes. These results highlight the importance of individualized approaches in tDCS treatments for dysphagia.

## 5. Conclusions

In conclusion, COMT Val158Met polymorphism may affect the plasticity of the mylohyoid motor cortex and actual swallowing function after tDCS. COMT polymorphism may be considered a potentially useful biomarker in the prognosis of neurostimulation as individualized treatment for dysphagia.

## Figures and Tables

**Figure 1 jpm-12-00488-f001:**
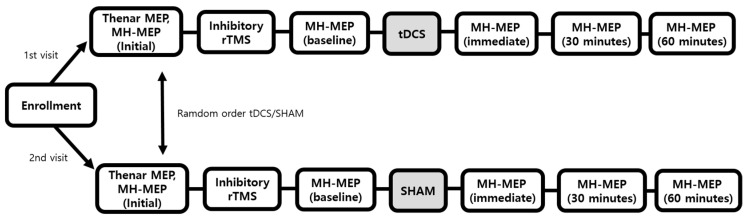
Flowchart of the participants. Abbreviation: MEP: motor-evoked potentials, MH-MEP: mylohyoid motor-evoked potentials, rTMS: repetitive transcranial magnetic stimulation, tDCS: transcranial direct current stimulation.

**Figure 2 jpm-12-00488-f002:**
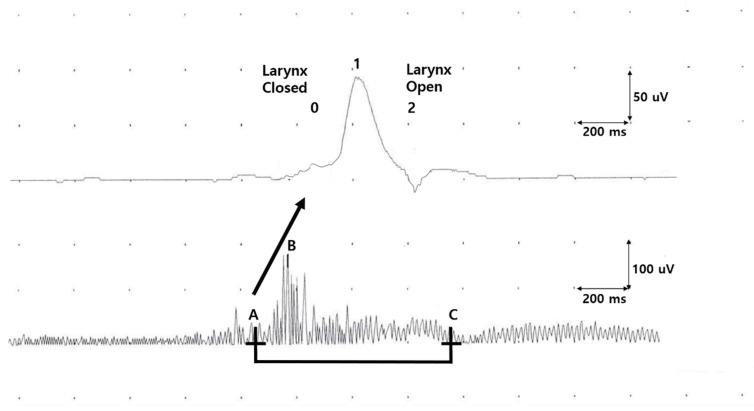
Evaluation of oropharyngeal swallowing function. An example of laryngeal movement signal (above) and submental electromyography (SM-EMG, below). “0” and “2” indicate the two subsequent deflections of laryngeal movement. “0” reflects the first deflection, which occurs with larynx and airway closure, and “2” reflects the second deflection with larynx opening. “1” indicates the point at which laryngeal movement reaches its maximum. “A” indicates the onset of SM-EMG signal, “B” indicates the peak amplitude of SM-EMG signal, and “C” indicates the end of SM-EMG signal. Based on these parameters, the laryngeal relocation time (“0”-“2” interval), triggering time (A-“0” interval), and total duration of SM-EMG (A–C interval) are depicted.

**Figure 3 jpm-12-00488-f003:**
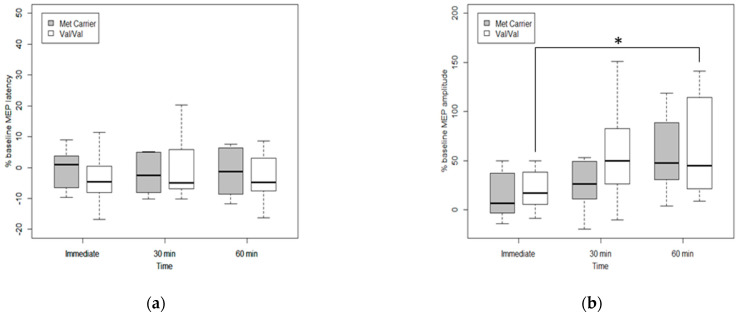
Cortical excitability changes across time according to COMT polymorphism. (**a**) Percentage changes of MH-MEP latencies and (**b**) MH-MEP amplitudes. The Val/Val group showed significant improvement MH-MEP amplitude percentage change across time from immediate to 60 min after tDCS (χ^2^ (2) = 11.17, *p*-value = 0.0038), confirmed by post hoc Wilcoxon signed-rank test and Bonferroni correction (adjusted *p*-value = 0.028). Statistically significant time intervals are indicated by a solid line and an asterisk (*).

**Figure 4 jpm-12-00488-f004:**
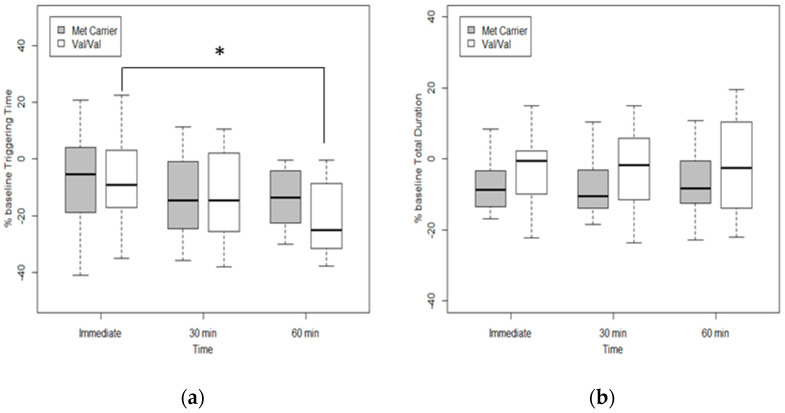
Swallowing function changes across time according to COMT polymorphism. (**a**) Percent changes of triggering time, (**b**) total duration, (**c**) maximal amplitude of submental-electromyography (SM-EMG), and (**d**) laryngeal relocation time from baseline. Friedman test was performed to assess changes across time of the response variables in Val/Val and Met carrier groups. Post hoc Wilcoxon signed-rank test in the Val/Val group showed statistically significant percentage changes from baseline in triggering time between immediate and 60 min after tDCS (adjusted *p*-value = 0.01). Statistically significant time interval differences are indicated by a solid line and an asterisk (*).

**Table 1 jpm-12-00488-t001:** Assessments of the participants according to the genotypes at the enrollment.

	Val/Val	Met Carriers	*p*-Value
No. of subjects	12	12	
Male (%)	7 (58.3%)	6 (50.0%)	1.000
APB rMT (mV)	20.4 ± 6.9	19.3 ± 3.2	0.628
tDCS session			
MH-MEP latency (ms)	4.7 ± 0.3	5.5 ± 0.5	0.209
MH-MEP amplitude (mV)	89.6 ± 21.9	98.8 ± 29.1	0.843
Triggering time (ms)	392.1 ± 47.0	406.9 ± 40.4	0.814
Total duration of SM-EMG (ms)	914.8 ± 69.4	953.6 ± 93.1	0.742
Maximal amplitude of SM-EMG (uV)	104.4 ± 5.7	117.5 ± 19.1	0.977
Laryngeal relocation time (ms)	712.7 ± 35.6	708.9 ± 38.3	0.944
Sham mode session			
MH-MEP latency (ms)	5.1 ± 0.3	5.7 ± 0.5	0.361
MH-MEP amplitude (mV)	89.5 ± 24.6	119.6 ± 30.0	0.478
Triggering time (ms)	391.6 ± 46.1	406.9 ± 40.4	0.805
Total duration of SM-EMG (ms)	915.6 ± 70.3	953.6 ± 93.1	0.748
Maximal amplitude of SM-EMG (uV)	103.9 ± 5.5	117.5 ± 19.1	0.977
Laryngeal relocation time (ms)	716.0 ± 35.1	708.9 ± 38.3	0.893

All measurements are expressed as mean values ± SEM (standard error of measurement). *p*-value < 0.05 was used for statistical significance. APB, abductor pollicis brevis; rMT, resting motor threshold; MH-MEP, mylohyoid motor-evoked potentials; SM-EMG, submental-electromyography.

**Table 2 jpm-12-00488-t002:** Comparison of tDCS and sham mode tDCS between post-inhibitory rTMS (baseline) and 60 min after intervention.

		Baseline	60 min	*p*-Value (Baseline vs. 60 min)	*p*-Value (tDCS vs. Sham Mode)
Cortical Excitability					
MH-MEP Latencies (ms)	tDCS	5.4 ± 0.3	5.3 ± 0.3	0.415	0.855
	Sham	5.6 ± 0.3	6.4 ± 0.9	0.376	
MH-MEP Amplitudes (mV)	tDCS	80.9 ± 17.6	152.6 ± 32.1	<0.001 *	0.014 *
	Sham	71.1 ± 14.6	86.2 ± 14.5	0.007 *	
Swallowing Function					
Triggering time (ms)	tDCS	453.0 ± 32.7	367.3 ± 24.7	<0.001 *	0.002 *
	Sham	426.3 ± 23.0	441.1 ± 31.2	0.627	
Total duration of SM-EMG (ms)	tDCS	1029.7 ± 62.0	992.4 ± 57.8	0.162	0.137
	Sham	981.8 ± 58.5	981.6 ± 55.0	0.52	
Maximal amplitude of SM-EMG (uV)	tDCS	110.7 ± 8.1	112.8 ± 10.8	0.617	0.440
	Sham	95.9 ± 6.6	104.6 ± 8.5	0.648	
Laryngeal relocation time (ms)	tDCS	813.0 ± 56.1	738.1 ± 44.1	0.032 *	0.074
	Sham	729.1 ± 30.1	741.6 ± 24.2	0.511	

All measurements are expressed as mean values ± SEM (standard error of measurement). * *p*-value < 0.05 was used for statistical significance. MH-MEP, mylohyoid motor-evoked potentials; SM-EMG, submental-electromyography.

## Data Availability

The datasets presented in this article are not readily available due to confidentiality agreements, and details of the data and how to request access should be directed to the corresponding authors.

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
