# Peer review of "Role of Catechol-O-methyltransferase Val158Met Polymorphism on Transcranial Direct Current Stimulation in Swallowing"

_jpm, 2022, doi:10.3390/jpm12030488_

Round 1
Reviewer 1 Report
In this manuscript, the authors investigate the role of the genetic variation of Catechol-O-methyltransferase (COMT) polymorphisms on transcranial direct current stimulation (tDCS) in swallowing. Overall, this work is interesting and well-presented. I would like to recommend the publication of this manuscript after addressing the following minor issues.
- The quality of Figure 2 should be improved.
- The statistical information should be well described in the figure captions of Figure 3 and Figure 4.
Author Response
Responses to Reviewers’ Comments
Manuscript Title: Role of Catechol-O-methyltransferase Val158Met Polymorphism on Transcranial Direct Current Stimulation in Swallowing
We sincerely appreciate you for your valuable remarks in this paper. We agree with the reviewer’s opinion and appreciate the constructive advice, which helped clarify and improve the manuscript considerably. Responses to the reviewers’ comments are listed below. We have highlighted each change within the revised manuscript.
We have prepared the responses in the attached manuscript.

Reviewer 2 Report
There is a lack of clarity throughout the paper that has made it very difficult to read and understand. There is an excessive amount of acronyms and abbreviations that confuses the reader to the point of not being able to understand the scope and experimentation of this study. I think the methods could be explained more clearly with a significantly less use of acronyms. I think there is great potential in this study with great scientific merit but the scientific findings need to be written more clearly with significantly less usage of acronyms and abbreviations. The acronyms that are absolutely needed in this paper should be included in a list.
Author Response
Responses to Reviewers’ Comments
Manuscript Title: Role of Catechol-O-methyltransferase Val158Met Polymorphism on Transcranial Direct Current Stimulation in Swallowing
We sincerely appreciate you for your valuable remarks in this paper. We agree with the reviewer’s opinion and appreciate the constructive advice, which helped clarify and improve the manuscript considerably. Responses to the reviewers’ comments are listed below. We have highlighted each change within the revised manuscript.
We have prepared the comments on the attached files.

Round 2
Reviewer 2 Report
The suggested changes are sufficient.